# A Novel Electrocardiographic Marker to Predict the Development of Preeclampsia: Frontal QRS-T Angle—A Prospective Pilot Study

**DOI:** 10.3390/medicina60111856

**Published:** 2024-11-12

**Authors:** Elif Uçar, Kenan Toprak, Mesut Karataş

**Affiliations:** 1Department of Gynecology and Obstetric, Private Esencan Hospital, Istanbul Esenyurt University, 63050 Istanbul, Turkey; eliflyy@hotmail.com; 2Department of Cardiology, Faculty of Medicine, Harran University, Viranşehir Road, Osmanbey Campus, 63300 Sanliurfa, Turkey; 3Kartal Koşuyolu High Specialization Training and Research Hospital, 63050 Istanbul, Turkey; mesut.cardio@gmail.com

**Keywords:** electrocardiography, Frontal QRS-T angle, preeclampsia

## Abstract

*Background and Objectives*: Preeclampsia, a pregnancy-induced hypertensive disorder, shares cardiovascular characteristics in etiology, prognosis, and fetomaternal risks. Electrocardiography plays a pivotal role in assessing cardiovascular risks. Beyond conventional predictors, identifying easily obtainable and reproducible electrocardiographic markers may significantly contribute to the early identification of individuals at risk of preeclampsia. In this study, we aimed to investigate the value of the Frontal QRS-T angle and other electrocardiographic parameters in predicting the development of preeclampsia. *Materials and Methods*: A total of 62 pregnant patients diagnosed with preeclampsia and 50 healthy pregnant patients as the control group were included in this study. The first- and third-trimester electrocardiographic parameters were compared within groups and between groups. *Results*: The Frontal QRS-T angle was significantly elevated in patients with preeclampsia compared to the controls (55.0 ± 40.8 vs. 19.5 ± 15.1; *p* = 0.002). The first-trimester Frontal QRS-T angles in the patients with preeclampsia were higher than those of the controls (29.5 ± 25.0 vs. 15.3 ± 11.5; *p* = 0.015). A high Frontal QRS-T angle independently marked preeclampsia development in antenatal and late pregnancy (*p* = 0.003 and *p* = 0.042, respectively). The diagnostic accuracy of the Frontal QRS-T angle in predicting preeclampsia surpassed other electrocardiographic parameters. *Conclusions*: This study shows that the Frontal QRS-T angle may be a candidate to be an independent predictor for the development of preeclampsia. In this context, the Frontal QRS-T angle, which is an electrocardiographic parameter, seems promising.

## 1. Introduction

Preeclampsia is seen in 2% to 8% of all pregnancies on average and remains a difficult disease to predict and treat; therefore, it is an important cause of fetomaternal mortality and morbidity [1]. Although the etiopathogenesis of preeclampsia has not been fully explained, it is suggested that the pathological feature develops by bioactive placental factors released into the circulation as a result of insufficient trophoblast invasion of the muscularis layer of the spiral arterioles at the 16th and 20th weeks of pregnancy [2,3]. The metabolic demands of the fetoplacental unit increase in the later stages of pregnancy. Insufficient invasion of spiral arterioles by trophoblasts inhibits the necessary expansion of spiral arterioles against this demand, which leads to placental dysfunction and manifests itself as preeclampsia [4]. The diagnosis of preeclampsia is classically made according to the criteria of new-onset hypertension (>140/90 mm Hg), new-onset proteinuria (defined as >300 mg/24 h or ≥2+ in the absence of a urinary tract infection in clear urinalysis), and edema in a pregnant or puerperal woman known to be normotensive before [1]. Preeclampsia is a pregnancy emergency and can lead to life-threatening complications, such as eclampsia, HELLP (hemolysis, elevated liver enzymes, low platelets) syndrome, kidney failure, disseminated intravascular coagulopathy (DIC), liver rupture, hypertensive crisis, pulmonary edema and hypertensive encephalopathy, and cortical blindness [4].

Since preeclampsia is associated with significant fetomaternal morbidity and mortality, it is essential to identify follow-up modalities that will require intensive measures according to early risk stratification in the management of these complications.

Eclampsia and preeclampsia are severe pregnancy-related cardiovascular conditions with substantial maternal and fetal morbidity risks. Identifying reliable prognostic factors for these conditions remains crucial for improving patient outcomes.

Although studies continue to investigate vasoactive mediators that play a role in the etiopathogenesis of preeclampsia and may be early indicators of preeclampsia, clinical practice still lacks the ability to predict patients at risk and treat the disease before it manifests [2,3,4,5,6].

It is known that preeclampsia is a syndrome rather than a uniform disease, and it has a heterogeneous structure in terms of presentation and pathophysiology, which may show different underlying phenotypes; its precursor effects actually begin before pregnancy, and the associated consequences, such as increased risk of cardiovascular disease development, extend beyond pregnancy. The increased risk of developing preeclampsia in women with cardiovascular risk factors indicates that pregnancy stress manifests the disease in those with subclinical vascular disease, rather than new-onset damage to intact vascular structure being caused by preeclampsia [6]. Studies have sought to develop risk models to identify patients at risk, mostly by looking for markers of injury or dysfunction in many affected organ systems [7]. Although many of them seem promising at first, they have not found application in clinical practice because they do not have the easy-to-apply, easily reproducible, cost-effective features that should be in a screening test [6,7,8]. The fact that preeclampsia screening is easily replicable and applicable will also provide the advantage of the ability to be used when developed preventive strategies are testable and likely to be effective.

Electrocardiography (ECG) is a reliable tool with proven efficacy that is used in daily practice to predict the risk of many cardiovascular diseases [9]. In recent years, a wealth of evidence has emerged regarding the prognostic utility of ECG intervals and waves beyond their diagnostic application in cardiovascular disease [10]. Electrocardiographic parameters, such as the P wave axis, duration and distribution, PR interval, Q wave, QRS and QTc intervals, QT dispersion, TpTe interval, and QRS-T angle, have been shown to be closely related to major adverse cardiovascular events (CVEs) and cardiac arrhythmias [9]. Thus, they have been proposed as prognostic markers for risk stratification in cardiovascular diseases [9].

Electrocardiographic studies in patients with preeclampsia are limited so far, and only ECGs during the development of preeclampsia (third trimester) were compared with healthy control groups matched with gestational age [11,12,13,14]. In this context, since the changes in the electrocardiographic parameters of preeclampsia cannot be compared with the ECG parameters before the disease appears, it is not known whether the ECG alterations change depending on the preeclamptic condition, which is a kind of acute cardiovascular process, and the early prognostic value of the electrocardiographic parameters cannot be determined.

Previous electrocardiographic studies on patients with preeclampsia did not examine the diagnostic and prognostic value of the Frontal QRS-T angle, and to the best of our knowledge, this study is the first to investigate the value of ECG parameters in predicting the development of preeclampsia. According to the results of this study, the Frontal QRS-T angle seems to be a promising electrocardiographic parameter that can be used to predict the development of preeclampsia early on.

## 2. Materials and Methods

### 2.1. Study Setting and Population

In this observational prospective study, 62 consecutive patients who developed preeclampsia and 50 consecutive healthy pregnant women who had healthy births, among 1246 pregnant women who applied to the outpatient clinic for routine examinations between June 2019 and December 2022, were included in this study. Both 1st-trimester and 3rd-trimester ECG data were obtained from 62 consecutive patients who developed preeclampsia and 50 consecutive healthy pregnant women matched with their gestational ages. Laboratory analysis of the study population was derived from whole blood parameters at the week of delivery. ECG parameters were compared both within the groups and between the groups. Diabetes, chronic hypertension, liver disease, chronic renal failure, electrolyte imbalance conditions, thromboembolic event or thrombophilic disease history, active infection, multiple pregnancies, pregnant women with HELLP syndrome, pregnant women using antiaggregants or anticoagulants, and subjects with missing data and incomplete or complete bundle branch block on ECG were excluded from the study.

This study was conducted in line with the principles of the Declaration of Helsinki. The study was approved by the local ethics committee (HRÜ/19.05.01; date: 27 March 2019). Informed consent was obtained from all participants.

### 2.2. Clinical Definitions

Preeclampsia was defined as the presence of one or more of the following new-onset conditions in pregnant women after 20 weeks of gestation: (1) proteinuria; (2) maternal organ dysfunction, including (a) renal failure (creatinine > 90 μmol/L; 1 mg/dL), (b) liver involvement (elevated transaminases with or without right upper quadrant or epigastric abdominal pain), (c) neurological complications (examples include eclampsia, altered mental status, blindness, stroke, hyperreflexia with clonus, severe headaches with hyperreflexia, and persistent visual scotomata), (d) hematological complications (thrombocytopenia with platelet count below 150,000/dL, disseminated intravascular coagulation, or hemolysis); and (3) uteroplacental dysfunction (such as fetal growth retardation or abnormal umbilical artery Doppler wave) [15]. Based on prior studies, we define the Frontal QRS-T angle operationally as the absolute difference between the QRS axis and T wave axis on the ECG. Specifically, the calculation is carried out using the following formula: Frontal QRS-T angle = |QRS axis − T axis|. Values above 180° are adjusted by subtracting from 360° to maintain a range between 0° and 180°. This standardized approach aligns with previously established methodologies in the literature [16].

### 2.3. Analysis of Electrocardiographic Parameters

In this study, 12-lead recorded ECG data taken at a 25 mm/s speed and a 10 mm/mV amplitude were used as standard. Parameters not automatically calculated using the ECG device were measured by 2 independent cardiologists using a digital caliper. ECG parameters obtained before starting antihypertensive medication were used in the study. QRS duration was recorded as the time from the beginning of the Q wave to the end of the S wave. The QT interval was measured from the beginning of the QRS complex to the end of the T wave. The QT and QTd intervals were corrected for heart rate (QTc; QTdc) using Bazett’s formula (QTc = QT/√RR) [16]. QTd was defined as the difference between the longest (QTmax) and shortest (QTmin) QT intervals from the 12 leads. The TpTe interval was defined as the interval from the peak of the T wave to the end of the T wave, and the longest interval measured from the 12 leads was recorded. Additionally, Tp-e/QT and Tp-e/QTc ratios were calculated. The S wave amplitude in lead V1 (SV1) and the R wave amplitude in lead V5 (RV5) were added to obtain [SV1 + RV5], which is a sign of cardiac hypertrophy and loading. The QRS and T wave axes automatically reported on the ECG recording were used to calculate the Frontal QRS-T angle using the following formula: Frontal QRS-T angle = |QRS axis − T axis| (Figure 1). When the angle was greater than 180°, the resulting value was subtracted from 360° for calculation, as previously reported [17]. The intraobserver and interobserver differences for the analyses were less than 5%.

### 2.4. Statistical Analysis

Statistical Program for Social Sciences 26 (IBM SPSS, Chicago, IL, USA) was used for statistical calculations. Kolmogorov–Smirnov test was used to determine whether the data fit the normal distribution. Continuous variables that fit the normal distribution were expressed as means ± standard deviation (SD), and those that did not fit the normal distribution were expressed as median with interquartile range (IQR). Comparisons between subjects with preeclampsia and the control group were analyzed using the Mann–Whitney U test and independent t-test where appropriate, and the same statistical methods were also used to compare preeclampsia subgroups. Chi-square test was applied to categorical variables. ECG parameters with statistical significance (*p* < 0.05) in univariate analysis were included in the multivariate logistic regression analysis. Multivariate regression analyses were performed to identify independent predictors of preeclampsia. Receiver operating characteristic (ROC) curve analysis was performed to determine the optimum cutoff value of the Frontal QRS-T angle and other electrocardiographic predictors for preeclampsia. Predictive validity was measured as the area under the ROC curve (c statistics), and comparisons of these parameters were made using MedCalc 16 statistical software (MedCalc Software Ltd., Ostend, Belgium). Given the study’s design, assuming a two-sample comparison with a significance level of 0.05, power of 0.80, and effect size based on the observed differences in Frontal QRS-T angle between the preeclampsia and control groups, a formal power analysis recommends a minimum sample size of 56 participants for the preeclampsia group and 46 for the control group, totaling 102 participants, which appears sufficient for detecting statistically significant differences in electrocardiographic parameters. The actual sample size in the study, with 62 participants in the preeclampsia group and 50 in the control group, exceeds these minimum requirements. All reported confidence interval (CI) values are calculated at the 95% level. Two-tailed *p* values < 0.05 were considered statistically significant.

## 3. Results

A total of 62 pregnant women with preeclampsia and 50 healthy pregnant women were included in this prospective observational study. There was no significant difference between the demographic, clinical, and biochemical parameters of the study groups (Table 1). The first- and third-trimester ECG parameters were compared within groups and between groups. When the first-trimester ECG parameters of the preeclampsia group and the control group were compared, the QTd and Frontal QRS-T angle were found to be significantly higher in the preeclampsia group than in the control group (28.1 ± 12.2 vs. 24.0 ± 7.4, *p* = 0.044 and 29.5 ± 25.0 vs. 15.3 ± 11.5, *p* = 0.015, respectively) (Table 2). When the third-trimester ECG parameters of the preeclampsia and control groups were compared, both the QTd and Frontal QRS-T angle were significantly higher in the preeclampsia group than in the control group (30.3 ± 11.0 vs. 26.9 ± 4.8, *p* = 0.046 and 55.0 ± 40.8 vs. 19.5 ± 15.1, *p* = 0.002, respectively) (Table 3). When the first- and third-trimester ECG parameters of the control group were compared among themselves, there was a significant difference only in terms of the heart rate, and the mean heart rate in the third trimester was significantly higher than the first trimester values (83.5 ± 11.6 vs. 69.8 ± 11.6; *p* < 0.001) (Table 4). When the first- and third-trimester ECG parameters of the preeclampsia group were compared among themselves, the QT interval was shorter (365.1 ± 43.3 vs. 387.1 ± 34.4; *p* = 0.012), and the heart rate (bpm) (86.5 ± 15.2 vs. 73.8 ± 10.4; *p* < 0.001), TpTe/QT ratio (0.21 ± 0.02 vs. 0.20 ± 0.02; *p* = 0.029), SV1 + RV5 (2.04 ± 0.6 vs. 1.76 ± 0.55; *p* = 0.010) and Frontal QRS-T angle (55.0 ± 40.8 vs. 29.5 ± 25.0, *p* = 0.008) parameters were significantly higher in the third trimester compared to the first trimester values (Table 5). While there was no difference between the first- and third-trimester Frontal QRS-T angle values in the control group (*p* = 0.073), there was a significant increase in the Frontal QRS-T angle in the third trimester compared to the first trimester in the preeclampsia group (*p* = 0.008) (Figure 2). In addition, the first-trimester Frontal QRS-T angle in the preeclampsia group was significantly higher than the first-trimester Frontal QRS-T angle of the control group (*p* = 0.015), and the third-trimester Frontal QRS-T angle in the preeclampsia group was significantly higher than the third-trimester Frontal QRS-T angle in the control group (*p* = 0.002) (Figure 2). In the multivariate logistic regression analysis, the first-trimester Frontal QRS-T angle (OR: 1.051, 95% CI: 1.017–1.086; *p* = 0.003), third-trimester Frontal QRS-T angle (OR: 1.013, 95% CI: 1.000–1.025; *p* = 0.042), and third-trimester [SV1 + RV5] (OR: 1.166, 95% CI: 1.011–1.278, *p* < 0.001) ECG parameters were determined as independent predictors for the development of preeclampsia (Figure 3). The receiver operating characteristic (ROC) curve analysis shows that the first-trimester Frontal QRS-T angle (QRS-T A1) can predict the development of preeclampsia with 63% sensitivity and 62% specificity at the best cut-off value of 17.5° (Area Under the Curve [AUC]: 0.695; 95% CI: 0.598–0.791; *p*< 0.001), and it shows that the third-trimester Frontal QRS-T angle (QRS-T A3) can predict the development of preeclampsia with 70% sensitivity and 68% specificity at the best cut-off value of 28.0° (AUC: 0.736; 95% CI: 0.641–0.832; *p* < 0.001) (Figure 4). The predictive values of the QTd1, QTd3, and [SV1 + RV5]3 parameters for preeclampsia were not statistically significant (*p* > 0.05, for all). When the ROC curves of the first- and third-trimester Frontal QRS-T angles underwent a pairwise comparison, there was no significant difference in their predictive power for preeclampsia (*p* = 0.508) (Figure 4).

QRS-T A1 indicates the 1st-trimester Frontal QRS-T angle; QRS-T A3 indicates the 3rd-trimester Frontal QRS-T angle; QTd1 indicates the 1st-trimester QT dispersion; QTd3 indicates the 3rd-trimester QT dispersion; and (SV1 + RV5)3 indicates the sum of the 3rd-trimester lead V1 S wave amplitude and lead V5 R wave amplitude.

## 4. Discussion

In this study, the value of ECG, which is an indispensable tool in the prediction, diagnosis, and follow-up of cardiovascular diseases in daily practice [9], was investigated in predicting preeclampsia, which is itself a cardiovascular disease risk factor [18], was i. The Frontal QRS-T angle and [SV1 + RV5], which are among the third-trimester ECG parameters, were determined as independent predictors of the development of preeclampsia, while only the Frontal QRS-T angle was determined as an independent predictor of the development of preeclampsia in the first trimester. In this context, the Frontal QRS-T angle appears as a unique ECG parameter in predicting the development of preeclampsia.

Preeclampsia is a pregnancy-specific, multisystem hypertensive disease and is among the most important causes of obstetric mortality and morbidity all over the world [1]. It is a disease generally characterized by an abnormal vascular response, manifested by an increase in systemic vascular resistance, increased platelet adhesion and aggregation, activation of the coagulation cascade, and endothelial cell dysfunction [19]. Preeclampsia is an important cause of obstetric mortality and morbidity today, and therefore, both early and accurate diagnosis are mandatory [7]. The identification of patients at risk for preeclampsia is conducive to increasing perinatal surveillance and possibly reducing maternal and fetal morbidity and mortality associated with preeclampsia [2,3,4,5]. Many methods, such as uterine artery Doppler ultrasound, the detection of angiogenic–antiangiogenic imbalance in maternal blood, and ophthalmic artery Doppler ultrasound, have been suggested to predict preeclampsia early on [2,3,4,5]. Although each of them has its own value, these methods are mostly not cost-effective, depend on the practitioner, and have low predictive values, so their applicability in the clinical field is low [7]. In order for a method to find clinical use as a kind of screening test in the clinical field, it must have features such as being reproducible, easy to apply, and cost-effective [20].

Today, ECG continues to be a reliable tool used in the diagnosis, treatment, and risk stratification of many diseases, especially cardiovascular diseases, in clinical practice [9,10]. The fact that ECG is a cost-effective, reproducible, and easily applicable tool has led to its widespread use in the clinical field. It has been shown in many clinical studies that it can guide the diagnosis and treatment modalities of cardiovascular diseases and reliably predict the risk of malignant arrhythmia [9,10].

In many clinical studies, it is well documented that those with preeclampsia have a significantly increased incidence of cardiovascular disease in late life [21]. In addition, it has been shown that conditions that are major risk factors for many cardiovascular diseases such as hypertension, obesity, hyperlipidemia, diabetes, and chronic kidney failure significantly increase the risk of preeclampsia in pregnant women [7,8]. In this context, the fact that major cardiovascular diseases increase the risk of developing preeclampsia and that preeclampsia is a risk factor for cardiovascular diseases in late life shows that preeclampsia is actually a kind of cardiovascular disease specific to pregnancy [21]. As a matter of fact, many clinical studies have clearly revealed this cause–effect-related cardiovascular risk scale of preeclampsia [22]. In this context, the predictive value of ECG for preeclampsia is important. Until now, various clinical studies have tried to reveal the differentiation of ECG parameters during normal pregnancy according to varying gestational periods [23,24]. In various clinical studies, ECG parameters were also examined in patients with preeclampsia, but in these studies, only ECG parameters during the current period were compared with the control group, and ECG parameters before disease development were not used in the study [12,13,14]. In the literature, only in a prospective study conducted by Angeli et al., only five patients developed preeclampsia and one patient developed eclampsia, and the Frontal-QRS-T angle was not evaluated in this study [25]. In the study conducted by Hoogsteder et al., electrocardiographic changes in women with a recent history of preeclampsia were examined, but the Frontal QRS-T angle and electrocardiographic parameters before preeclampsia development were not evaluated [26]. This deprived these studies of the ability to identify an ECG parameter that could predict the development of preeclampsia because it is not known whether the change in electrocardiographic parameters is newly developed due to an acute condition due to preeclampsia or is a demonstration of an already existing pro-messenger.

Cardiovascular diseases are a major determinant of adverse events, such as malignant cardiac arrhythmias and sudden cardiac death (SCD), and heterogeneity in myocardial repolarization plays a pivotal role in the development of cardiac arrhythmias and SCD [27]. The Frontal QRS-T angle, which is a novel electrocardiographic marker of the heterogeneity of myocardial repolarization, is described as the absolute difference between myocardial depolarization (QRS axis) and repolarization (T axis) [27]. The Frontal QRS-T angle can be easily calculated using a 12-lead superficial ECG; its abnormality reflects the electrical instability of the myocardium, and the clinical significance of its high values in many cardiovascular diseases is well documented [28,29]. In various clinical studies, ventricular repolarization indexes such as the QTd and TpTe/QT were found to be significantly higher in patients with preeclampsia compared to the healthy control group, but the diagnostic and predictive value of the Frontal QRS-T angle was not evaluated in these studies [13,30].

In our study, the Frontal QRS-T angle and QTd values were higher in patients with preeclampsia compared to the control group matched with gestational age. Interestingly, these findings were consistent with the ECG parameters obtained at the first antenatal examination, that is, the Frontal QRS-T angle and QTd were significantly higher in patients with preeclampsia compared to the control group, even in the first trimester. Moreover, the Frontal QRS-T angle and QTd values obtained at the first antenatal examination in the control group did not differ significantly with the third-trimester values, while the Frontal QRS-T angle obtained during the preeclampsia period increased significantly in the preeclampsia group, and no significant change was observed in QTd. These results point to the potential of the Frontal QRS-T angle to be an electrocardiographic marker that can predict preeclampsia early on. In particular, the Frontal QRS-T angle is an indicator of heterogeneity in ventricular activity, and its high values are associated with the incidence of ventricular arrhythmias and are indicative of all-cause mortality [27,28,29,30,31]. In this context, high Frontal QRS-T angle values may explain the increased incidence of severe ventricular arrhythmias and adverse outcomes in pregnant women with preeclampsia [22].

Another interesting finding of our study is that the [SV1 + RV5] parameter, which is a sign of ventricular loading, increased significantly in patients with preeclampsia compared to the first trimester, while it did not show significant variability in the healthy control group. Physiological increases in heart rate, cardiac output, and intravascular volume are observed during pregnancy [30]. This adaptive process facilitates the adaptation of the cardiovascular system to the increased metabolic needs of the mother, ensuring adequate delivery of oxygenated blood to peripheral tissues and the fetus. In the course of preeclampsia, however, this reverts to a maladaptive process to exacerbation, possibly resulting in an increase in the QRS voltage parameter, an electrocardiographic sign of cardiac overload.

The scientific hypothesis positing a relationship between preeclampsia and the Frontal QRS-T angle revolves around the premise that preeclampsia, a pregnancy-related hypertensive disorder, induces cardiovascular changes that may reflect in ECG markers. Preeclampsia is known to cause endothelial dysfunction, increased systemic vascular resistance, and altered cardiac loading conditions, which can lead to left ventricular hypertrophy and strain [31,32]. These changes may impact the depolarization and repolarization processes of the heart, leading to an increased Frontal QRS-T angle in ECG. The QRS-T angle, representing the spatial difference between ventricular depolarization and repolarization vectors, is considered a marker of cardiac electrical heterogeneity and has been associated with an increased risk of arrhythmias and adverse cardiovascular events. Therefore, a widened QRS-T angle in preeclampsia may reflect underlying subclinical cardiac dysfunction, which could contribute to the increased cardiovascular risk observed in women with this condition.

## 5. Limitations

The main limitation of our study is that it included a relatively small study population. Due to the cross-sectional design, bias cannot be completely ruled out, and unknown confounders may have influenced the results, although known risk factors were added to the multivariate regression analysis to identify independent predictors of preeclampsia. In addition, due to the cross-sectional and observational design, the cause–effect relationship between the Frontal QRS-T angle and preeclampsia cannot be established clearly, and the results cannot be generalized to all patients with preeclampsia due to the small sample size. In addition, the study population did not have serial ECG follow-ups. Also, the effect of the Frontal QRS-T angle on long-term clinical outcomes is not clearly known since the study population does not have long-term follow-up results. Randomized clinical trials involving larger study populations are essential to confirm our study results and to establish a cut-off value for a Frontal QRS-T angle that may predict preeclampsia.

## 6. Conclusions

This study highlights the potential of the Frontal QRS-T angle as an early, non-invasive ECG marker for predicting preeclampsia, addressing a critical need for accessible early detection tools. A widened Frontal QRS-T angle in the first trimester was associated with the later development of preeclampsia, suggesting that this cost-effective and easy-to-calculate parameter could be integrated into routine prenatal care. Further research with larger cohorts is needed to confirm these findings and establish standardized cut-off values for clinical use.

## Figures and Tables

**Figure 1 medicina-60-01856-f001:**
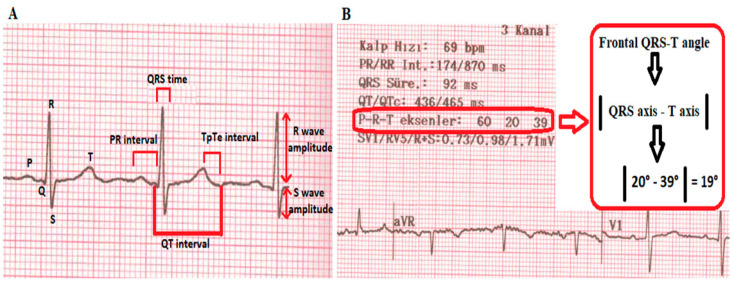
(**A**) A display of the electrocardiographic parameters on the ECG recording. (**B**) A simple calculation of the Frontal QRS-T angle from the ECG recording. The Frontal QRS-T angle, also referred to as the QRS-T axis difference or angle, is a measure of the absolute difference between the QRS axis and T wave axis on an ECG. It is calculated using the formula |QRS axis − T axis|. For angles greater than 180°, the value is adjusted by subtracting from 360° to keep the measurement within 0° to 180°.

**Figure 2 medicina-60-01856-f002:**
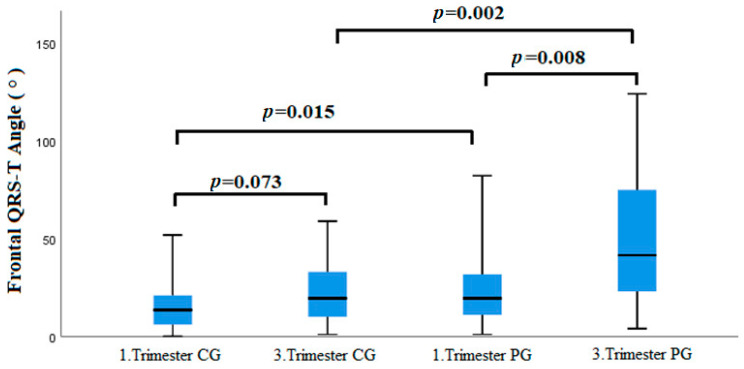
A comparison of the Frontal QRS-T angle values in the 1st and 3rd trimesters between the control group and the preeclampsia group. CG, control group; PG: preeclampsia group.

**Figure 3 medicina-60-01856-f003:**
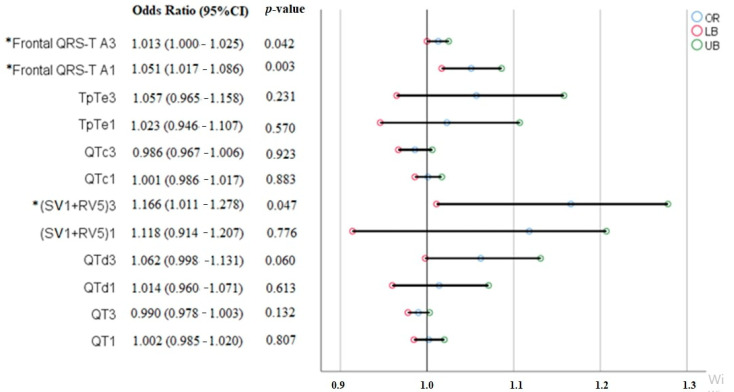
Independent electrocardiographic predictors (*) of preeclampsia: Frontal QRS-T A3, Frontal QRS-T A1, and (SV1 + RV5)3. The numbers (1 or 3) next to the ECG parameters indicate which trimester the parameter belongs to. CI: Confidence Interval; OR: Odds Ratio; LB: Lower Band; UB: Upper Band.

**Figure 4 medicina-60-01856-f004:**
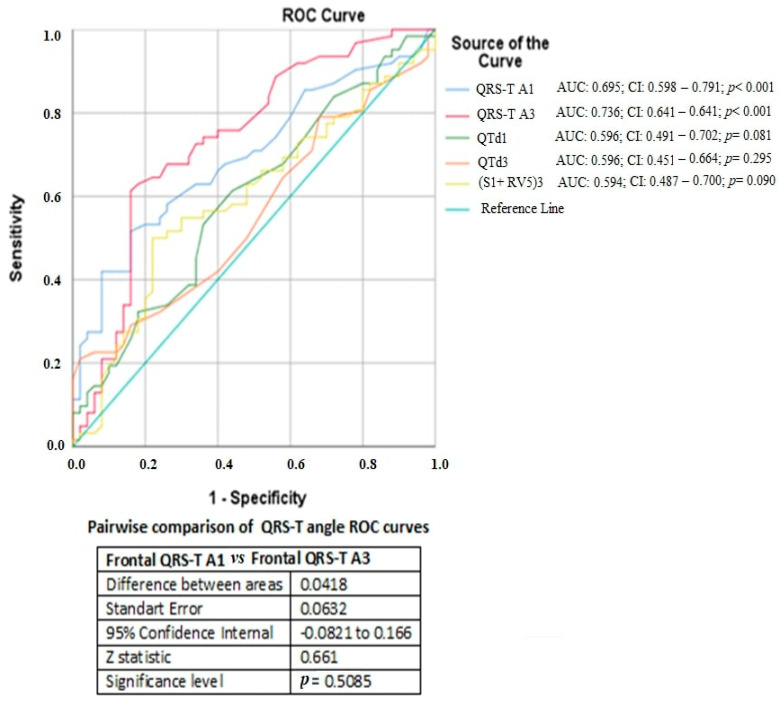
The ROC curve analysis shows that the 1st-trimester Frontal QRS-T angle (QRS-T A1) can predict the development of preeclampsia with 63% sensitivity and 62% specificity at the best cut-off value of 17.5°(AUC, 0.695; 95% CI, 0.598–0.791; *p* < 0.001), and it shows that the 3rd-trimester Frontal QRS-T angle (QRS-T A3) can predict the development of preeclampsia with 70% sensitivity and 68% specificity at the best cut-off value of 28.0°(AUC, 0.736; 95% CI, 0.641–0.832; *p* < 0.001). The preeclampsia predictive values of the QTd1, QTd3, and (SV1 + RV5)3 parameters were not statistically significant (*p* > 0.05 for all). When the ROC curves of the 1st- and 3rd-trimester Frontal QRS-T angles underwent a pairwise comparison, there was no significant difference in their predictive power for preeclampsia (*p* = 0.508).

**Table 1 medicina-60-01856-t001:** A comparison of the demographic characteristics and hematological and biochemical parameters of the study groups.

Variables	Total (*n* = 112)	Control Group (*n* = 50)	Preeclampsia Group (*n* = 62)	*p*-Values
Age, years	32.2 ± 5.2	32.0 ± 6.2	32.3 ± 4.4	0.813
Weight (kg)	79.6 ± 10.7	79.94± 7.6	79.4 ± 12.7	0.632
Height (cm)	162.9 ± 3.3	162.7 ± 3.3	163.1 ± 3.4	0.796
BMI (kg/m^2^)	29.5 ± 3.6	30.0 ± 2.5	29.0 ± 4.3	0.175
Gravidity (n)	4.7 ± 2.7	4.7 ± 3.1	4.8 ± 2.4	0.846
Parity (n)	3.3 ± 2.7	3.3 ± 2.6	3.2 ± 2.9	0.886
SBP (mmHg)	163 (145–196)	122 (118–126)	161 (154–175)	<0.001
DBP (mmHg)	98 (83–124)	76 (73–79)	97 (93–106)	<0.001
MAP (mmHg)	148 (106–119)	91 (89–93)	120 (114.6–128)	<0.001
Respiratory rate (n)	24.6 ± 3.3	24.5 ± 3.7	24.7 ± 2.9	0.780
Hemoglobin (g/dL)	11.15 ± 0.54	11.53 ± 0.44	10.89 ± 0.61	0.313
Hematocrit (%)	35.9 (33.9–37.0)	36.5 (34.1–36.9)	35.6 (33.7–37.1)	0.152
BUN (mg/dL)	36.8 (31.2 ± 43.4)	37.3 (33.4–43.6)	36.4 (28.1–43.8)	0.101
Uric acid (mg/dL)	5.3 (4.3 ± 6.0)	5.3 (4.2–5.8)	5.4 (4.1–6.1)	0.255
Creatinine (mg/dL)	0.7 (0.6–1.0)	0.7 (0.6–0.9)	0.7 (0.6–1.0)	0.114
ALT (U/L)	23.8 (19.9- 34.1)	24.3 (21.0–33.0)	23.5 (19.6–34.3)	0.147
AST (U/L)	33.1 (21.2–41.9)	32.0 (21.6–41.3)	34.0 (21.3–42.4)	0.226
WBC (x1000/mm^3^)	6.78 (4.36–7.28)	6.71 (4.29–7.20)	6.80 (4.48–7.45)	0.138
CRP (mg/dL)	0.50 (0.21–0.97)	0.48 (0.19–0.96)	0.51 (0.22–1.10)	0.096
Platelet count (×1000/mm^3^)	288.4 ± 95.7	286.2 ± 83.4	291.7 ± 120.7	0.654
Triglycerides (mg/dL)	265.7 ± 91.3	237.9 ± 70.6	259.1 ± 81.3	0.113
Total cholesterol (mg/dL)	218.4 ± 86.8	200.0 ± 51.6	204.4 ± 64.6	0.137
HDL-C (mg/dL)	47.1 ± 15.5	40.8 ± 13.4	42.3 ± 15.4	0.315
LDL-C (mg/dL)	99.3 ± 37.1	99.5 ± 37.6	98.7 ± 37.1	0.912
Obstetric follow-up results				
Newborn’s birth weight (g)	2748 ± 401	2837 ± 456	2652 ± 384	0.134
Gestational age at delivery	37 (36–38)	37 (36–39)	37 (35–38)	0.125

Values are shown as mean ± SD, n (%), or median (interquartile range) unless otherwise stated. Abbreviations: BMI: body mass index; SBP: systolic blood pressure; DBP: diastolic blood pressure; MAP mean arterial pressure; BUN: blood urea nitrogen; AST: aspartate aminotransferase; ALT: alanine aminotransferase; WBC: white blood cell; CRP: C-reactive protein; HDL-C: high-density lipoprotein cholesterol; LDL-C: low-density lipoprotein cholesterol.

**Table 2 medicina-60-01856-t002:** Comparison of 1st-trimester ECG parameters of patients with preeclampsia and control group.

ECG Parameters	Control Group (*n* = 50)	Preeclampsia Group (*n* = 62)	*p*-Value
	1st Trimester	1st Trimester	
Heart rate, bpm	69.8 ± 11.6	73.8 ± 10.4	0.057
PR interval (ms)	144.8 ± 18.0	146.2 ± 22.4	0.711
QRS time (ms)	87.7 ± 11.1	87.9 ± 9.6	0.942
QT interval (ms)	390.5 ± 23.8	387.1 ± 34.4	0.556
QTc interval (ms)	421.8 ± 23.6	419.0 ± 56.5	0.744
QTd (ms)	24.0 ± 7.4	28.1 ± 12.2	0.044
TpTe interval (ms)	77.2 ± 5.1	77.7 ± 7.5	0.648
TpTe/QT	0.19 ± 0.01	0.20 ± 0.02	0.305
TpTe/QTc	0.18 ± 0.01	0.20 ± 0.02	0.382
SV1 + RV5 (mV)	1.74 ± 0.65	1.76 ± 0.55	0.810
Frontal QRS-T angle (°)	15.3 ± 11.5	29.5 ± 25.0	0.015

Values are expressed as mean ± SD.

**Table 3 medicina-60-01856-t003:** Comparison of 3rd-trimester ECG parameters of patients with preeclampsia and control group.

ECG Parameters	Control Group (*n* = 50)	Preeclampsia Group (*n* = 62)	*p*-Value
	3rd Trimester	3rd Trimester	
Heart rate, bpm	83.5 ± 11.6	86.5 ± 15.2	0.264
PR interval (ms)	136.08 ± 22.7	139.1 ± 20.3	0.994
QRS time (ms)	88.5 ± 10.5	90.2 ± 10.6	0.399
QT interval (ms)	371.5 ± 29.5	365.1 ± 43.3	0.376
QTc interval (ms)	415.2± 46.4	432.4 ± 37.2	0.691
QTd (ms)	26.9 ± 4.8	30.3 ± 11.0	0.046
TpTe interval (ms)	75.5 ± 3.9	76.1 ± 6.7	0.590
TpTe/QT	0.20 ± 0.02	0.21 ± 0.02	0.082
TpTe/QTc	0.19 ± 0.01	0.17 ± 0.01	0.327
SV1 + RV5 (mV)	1.85 ± 0.55	2.04 ± 0.62	0.093
Frontal QRS-T angle (°)	19.5 ± 15.1	55.0 ± 40.8	0.002

Values are expressed as mean ± SD.

**Table 4 medicina-60-01856-t004:** A comparison of the 1st- and 3rd-trimester ECG parameters of the control group.

ECG Parameters	Control Group (*n* = 50)		*p*-Value
	1st Trimester	3rd Trimester	
Heart rate, bpm	69.8 ± 11.6	83.5 ± 11.6	<0.001
PR interval (ms)	144.8 ± 18.0	136.08 ± 22.7	0.051
QRS time (ms)	87.7 ± 11.1	88.5 ± 10.5	0.726
QT interval (ms)	390.5 ± 23.8	371.5 ± 29.5	0.198
QTc interval (ms)	421.8 ± 23.6	415.2± 46.4	0.097
QTd (ms)	24.0 ± 7.4	26.9 ± 4.8	0.070
TpTe interval (ms)	77.2 ± 5.1	75.5 ± 3.9	0.077
TpTe/QT	0.19 ± 0.01	0.20 ± 0.02	0.220
TpTe/QTc	0.18 ± 0.01	0.19 ± 0.01	0.100
SV1 + RV5 (mV)	1.74 ± 0.65	1.85 ± 0.55	0.342
Frontal QRS-T angle (°)	15.3 ± 11.5	19.5 ± 15.1	0.073

Values are expressed as mean ± SD.

**Table 5 medicina-60-01856-t005:** A comparison of the 1st- and 3rd-trimester ECG parameters of the preeclampsia group.

ECG Parameters	Preeclampsia Group (*n* = 62)		*p*-Value
	1st Trimester	3rd Trimester	
Heart rate, bpm	73.8 ± 10.4	86.5 ± 15.2	<0.001
PR interval (ms)	146.2 ± 22.4	139.1 ± 20.3	0.066
QRS time (ms)	87.9 ± 9.6	90.2 ± 10.6	0.206
QT interval (ms)	387.1 ± 34.4	365.1 ± 43.3	0.012
QTc interval (ms)	419.0 ± 56.5	432.4 ± 37.2	0.123
QTd (ms)	28.1 ± 12.2	30.3 ± 11.0	0.290
TpTe interval (ms)	77.7 ± 7.5	76.1 ± 6.7	0.207
TpTe/QT	0.20 ± 0.02	0.21 ± 0.02	0.029
TpTe/QTc	0.20 ± 0.02	0.17 ± 0.01	0.109
SV1 + RV5 (mV)	1.76 ± 0.55	2.04 ± 0.62	0.010
Frontal QRS-T angle (°)	29.5 ± 25.0	55.0 ± 40.8	0.008

Values are expressed as mean ± SD.

## Data Availability

The data that support the findings of this study are available upon request from the corresponding author. The data are not publicly available due to privacy or ethical restrictions.

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
