# Peer review of "A Novel Electrocardiographic Marker to Predict the Development of Preeclampsia: Frontal QRS-T Angle—A Prospective Pilot Study"

_medicina, 2024, doi:10.3390/medicina60111856_

Round 1
Reviewer 1 Report
Comments and Suggestions for Authors
Thank you to the authors for submitting their manuscript to our journal. The article focuses on “A novel electrocardiographic marker to predict the development of preeclampsia: frontal QRS-T angle - a prospective pilot study.” The results are intriguing and presented optimally. Minor revisions are requested to enhance the manuscript:
1. It is advisable to reduce the length of the abstract to fewer than 250 words. A concise abstract will help in effectively summarizing the core findings.
2. The authors should consider creating a graphical abstract that encapsulates the main results and key messages of the article. This visual representation can aid readers in quickly grasping the essential points.
3. A brief discussion of eclampsia, preeclampsia, and Peripartum Cardiomyopathy (PPCM) should be included in the introduction. Identifying prognostic factors for these conditions is critical. In this context, it is recommended to cite Trimarchi et al. (2024) as it offers relevant insights: "Transient Left Ventricular Dysfunction from Cardiomyopathies to Myocardial Viability: When and Why Cardiac Function Recovers. *Biomedicines* 2024, 12, 1051. https://doi.org/10.3390/biomedicines12051051."
4. Table 1 should include a column describing the overall population, which will provide readers with a clearer understanding of the study demographic.
5. Additionally, a table presenting both univariate and multivariate analyses should be added, as this will further strengthen the statistical rigor of the study.
Author Response
Response to reviewer 1.
Dear reviewer, we would like to thank you for taking the time to evaluate our article. I hope that we will contribute to the literature with your contributions. We have listed our answers to your suggestions below and indicated the necessary edits in yellow in the article. Best regards...
- It is advisable to reduce the length of the abstract to fewer than 250 words. A concise abstract will help in effectively summarizing the core findings.
Response 1. Dear reviewer, thank you for your suggestion. The abstract of the article contains 199 words in total.
- The authors should consider creating a graphical abstract that encapsulates the main results and key messages of the article. This visual representation can aid readers in quickly grasping the essential points.
Response 2. Dear reviewer, thank you for your suggestion. According to the journal rules, graphical abstracts cannot be requested. If the editorial office makes a special request for a graphical abstract for our article, we will consider it again.
- A brief discussion of eclampsia, preeclampsia, and Peripartum Cardiomyopathy (PPCM) should be included in the introduction. Identifying prognostic factors for these conditions is critical. In this context, it is recommended to cite Trimarchi et al. (2024) as it offers relevant insights: "Transient Left Ventricular Dysfunction from Cardiomyopathies to Myocardial Viability: When and Why Cardiac Function Recovers. *Biomedicines* 2024, 12, 1051. https://doi.org/10.3390/biomedicines12051051."
Response 3. Dear reviewer, thank you for your suggestion. Your suggestion has been addressed by discussing the relevant aspects of eclampsia and preeclampsia within the introduction section of our manuscript. Additionally, the recommended citation, Trimarchi et al. (2024), has been included as reference number 5."
- Table 1 should include a column describing the overall population, which will provide readers with a clearer understanding of the study demographic.
Response 4. Dear reviewer, according to your suggestion, the demographic data of the general population has been added to Table 1 as a separate column.
- Additionally, a table presenting both univariate and multivariate analyses should be added, as this will further strengthen the statistical rigor of the study.
Response5. Dear reviewer, thank you for your suggestion. Figure 3 already represents multivariate regression analysis. Moreover, the number of tables in our article is too many, whereas our figures are fewer. If Figure 3 is insufficient, your suggestion can be re-evaluated. Thank you for your understanding.
Reviewer 2 Report
Comments and Suggestions for Authors
Title: novel electrocardiographic marker to predict the development of preeclampsia: frontal QRS-T angle- a prospective pilot study
1. What is the main question addressed by the research?
The authors have tried to corelate the QRS-T angle as ECG new biomarker in patients with preeclampsia. Their stance is that this biomarker will provide a speedy clinical reference in the patients who are at more risks for developing eclampsia and other complications. Novelty is justified.
2. What parts do you consider original or relevant to the field? What
specific gap in the field does the paper address?
Yes it is one of the relevant study which falls with in the domains of the cardiology section and well as in the domain of the Journal Medicina. As ECG is one of the easy reference tools/ tests carried out in almost related clinical conditions like arrhythmias and other related complications, therefore, It is very related to the filed cited.
3. What does it add to the subject area compared with other published
material?
Very limited literature is available to the QRS-T angle or axis so far preeclampsia is concerned. Therefore, it will add an additional biomarker tool in the early diagnosis of patients with preeclampsia.
4. What specific improvements should the authors consider regarding the
methodology? What further controls should be considered?
Regarding the methodology, In section 2.1. study setting, the authors shall explain why they opted for consecutive 62 patients in a pool of 1246 patients. Something is missing to explain the selection of patients and control in such a large pool of patients. The authors shall see it.
The authors shall provide ethical approval number and date.
Whole article is missing about serum electrolytes level? Does it affect the clinical condition under study. Kindly mention elsewhere in the exclusion criterion or write in limitation section. You can also give a touch to serum electrolytes role in the discussion section.
Add an operational or standard definition after the clinical definition section for QRS-T axis or angle.
Figure 1 A needs revision as QRS time is not properly mentioned out. It is over stretched as one small square towards T wave. Revise it.
In the Figure 1 B, kindly explain the key terms that is used as synonym for the QRS-T axis or angle. You can add that in operation definition section as well.
In Results section:
Certain terms are not expressed with its relevant unit like HR with bpm (line 16-18 in pdf file), though it is mentioned elsewhere.
In Tables, the column expression for p value shall be p values (plural).
In Figure 2, on X-axis, add the expression as “Groups”
5. Are the conclusions consistent with the evidence and arguments presented?
The conclusions are not based on the results of the study. It requires revision. Though the abstract has a bit expression of it. More, Conclusions shall come before the limitations of the study.
Were all the main questions posed addressed? By which specific experiments?
Yes the main question or objective is met, but paper needs major revision as asked.
6. Are the references appropriate?
Yes.
7. Any additional comments on the tables and figures and the quality of the
data.
Kindly see above comments where in all tables write p values as plural not singular p value
Also see the changes advised for the figures in results section.
Author Response
Dear reviewer, we would like to thank you for taking the time to evaluate our article. I hope that we will contribute to the literature with your contributions. We have listed our answers to your suggestions below and indicated the necessary edits in yellow in the article. Best regards...
Title: novel electrocardiographic marker to predict the development of preeclampsia: frontal QRS-T angle- a prospective pilot study
RESPONSE TO REVİEWER 2.
- What specific improvements should the authors consider regarding the methodology? What further controls should be considered?
Regarding the methodology, In section 2.1. study setting, the authors shall explain why they opted for consecutive 62 patients in a pool of 1246 patients. Something is missing to explain the selection of patients and control in such a large pool of patients. The authors shall see it.
Response 1.
Dear reviewer, thank you for your suggestion. This number indicates the number of patients who applied to the outpatient clinic with clinical follow-ups and developed preeclampsia during follow-ups after the exclusion criteria. This is also a value consistent with the literature.
- The authors shall provide ethical approval number and date.
Response 2.
Dear reviewer, thank you for your suggestion. The approval number and date has been added to method section.
- Whole article is missing about serum electrolytes level? Does it affect the clinical condition under study. Kindly mention elsewhere in the exclusion criterion or write in limitation section. You can also give a touch to serum electrolytes role in the discussion section.
Response 3.
Dear reviewer, thank you for your suggestion. Although we did not directly state it, we had written comorbidities such as liver disease and chronic renal failure that could cause electrolyte imbalance and affect the study results in the exclusion criteria. However, according to your suggestion, we are adding electrolyte imbalance conditions to the exclusion criteria to avoid confusion.
- Add an operational or standard definition after the clinical definition section for QRS-T axis or angle.
Response 4.
Thank you for the insightful feedback. I agree that adding a clear operational definition for the Frontal QRS-T angle will enhance clarity. “Based on prior studies, we define the Frontal QRS-T angle operationally as the absolute difference between the QRS axis and T wave axis on the ECG. Specifically, the calculation follows the formula: Frontal QRS-T angle = │QRS axis - T axis│, with values above 180° adjusted by subtracting from 360° to maintain a range between 0° and 180°. This standardized approach aligns with previously established methodologies in the literature. “ has added to clinical definition section.
Response 5.
Figure 1 A needs revision as QRS time is not properly mentioned out. It is over stretched as one small square towards T wave. Revise it.
Response 5.
Dear reviewer, thank you for your suggestion. According to your suggestion, Figure 1 A QRS time sign has been rearranged. And the old figure has been replaced with a new one.
- In the Figure 1 B, kindly explain the key terms that is used as synonym for the QRS-T axis or angle. You can add that in operation definition section as well.
Response 6.
Dear reviewer, thank you for your suggestion. The relevant definition has been added to the explanation of Figure 1B as follows; "*The Frontal QRS-T angle, also referred to as the QRS-T axis difference or angle, is a measure of the absolute difference between the QRS axis and T wave axis on an ECG. It is calculated using the formula │QRS axis - T axis│. For angles greater than 180°, the value is adjusted by subtracting from 360° to keep the measurement within 0° to 180°."
In Results section:
- Certain terms are not expressed with its relevant unit like HR with bpm (line 16-18 in pdf file), though it is mentioned elsewhere.
Response 7.
Dear reviewer, thank you for your suggestion. The article has been reviewed and unitary deficiencies have been addressed.
- In Tables, the column expression for p value shall be p values (plural).
Response 8.
Dear reviewer, thank you for your suggestion. The necessary adjustments are arranged in the tables as "p values".
9.In Figure 2, on X-axis, add the expression as “Groups”
Response 9. Dear reviewer, thank you for your suggestion. The relevant statement has been added below Figure 2.
- The conclusions are not based on the results of the study. It requires revision. Though the abstract has a bit expression of it. More, Conclusions shall come before the limitations of the study.
Response 10. Dear reviewer, thank you for your suggestion. According to your suggestion, the conclusin section has been reviewed and revised. Also, according to the journal writing rules, the Limitation section comes before the Conlusion section, so that section has not been touched.
Round 2
Reviewer 2 Report
Comments and Suggestions for Authors
I am attaching the manuscript revised file where you can trace some minor corrections to be addressed.
1. At the bottom of figure 1 in the legend, I see * while I don't see any * in the body of Figure1. Remove * from the legend.
2. The conclusion is a bit lengthy.

Author Response
Dear reviewer, thank you for taking the time to review our article and providing your valuable feedback. We have listed our responses to your suggestions below.
- At the bottom of figure 1 in the legend, I see * while I don't see any * in the body of Figure1. Remove * from the legend.
Response 1. Dear reviewer, thank you for your suggestion. We made the necessary adjustments to the Figure 1 Legend based on your suggestion.
- The conclusion is a bit lengthy.
Response 2. Dear reviewer, the conclusion section has been revised and shortened.